# Physical Activity during the COVID-19 Pandemic in the UK: A Qualitative Analysis of Free-Text Survey Data

**DOI:** 10.3390/ijerph192214784

**Published:** 2022-11-10

**Authors:** Verity Hailey, Alexandra Burton, Mark Hamer, Daisy Fancourt, Abigail Fisher

**Affiliations:** 1Department of Behavioural Science and Health, University College London, London WC1E 6BT, UK; 2Institute Sport Exercise & Health, Division Surgery & Interventional Science, University College London, London W1T 7HA, UK

**Keywords:** COVID-19, lockdown, physical activity, mental health, wellbeing, outdoor space

## Abstract

Background: Several quantitative studies have found a decline in physical activity in response to COVID-19 pandemic restrictions. The aim of the present study was to use large-scale free text survey data to qualitatively gain a more in-depth understanding of the impact of the COVID-19 pandemic on physical activity, then map barriers and facilitators to the Capability, Opportunity, Motivation, and Behaviour (COM-B) Model of Behaviour to aid future intervention development. Methods: 17,082 participants provided a response to the free text module, and data from those who mentioned a physical activity related word in any context were included. Data were analysed using thematic analysis and key themes identified. Results: 5396 participants provided 7490 quotes related to physical activity. The sample were predominately female (84%), white (British/Irish/Other) (97%) and aged <60 years (57%). Seven key themes were identified: the importance of outdoor space, changes in daily routine, COVID-19 restrictions prevented participation, perceived risks or threats to participation, the importance of physical health, the importance of physical activity for mental health and the use of technology. Conclusion: Future physical activity interventions could encourage people to walk outdoors, which is low cost, flexible, and accessible to many. Developing online resources to promote and support physical activity provides a flexible way to deliver quality content to a large audience.

## 1. Background

The COVID-19 pandemic had a profound impact on society with measures taken to control the virus, such as lockdowns, travel restrictions and social distancing fundamentally changing the way we work, socialise and exercise [1]. It is well established that sufficient physical activity is important for prevention of many chronic diseases [2,3]. This may be particularly important during a pandemic as demonstrated by a nationwide cohort study (South Korea) and a retrospective observational study (USA) which found that active adults had reduced risk of severe infection by COVID-19 [4,5]. Adults who consistently met physical activity guidelines had a reduced risk of hospitalisation, admission to critical care and death compared to those who were consistently inactive [5]. A case–control study (Sweden) emphasised the importance of maintaining or increasing cardiovascular fitness to strengthen resilience to severe COVID-19 [6]. Studies also found physical inactivity and weight-related comorbidities were significant risk factors for contracting COVID-19 [7]. In non-pandemic settings, a meta-analysis of 54 studies (7 observational, 42 interventional, 5 vaccination studies), looked at the effect of regular physical activity on the immune system and risk of community acquired infection. This found regular moderate to vigorous physical activity increases resistance to infectious disease, reducing the risk of community acquired infection, and infectious disease mortality in the general population [8].

A large number of quantitative studies demonstrated that physical activity substantially reduced as a result of the COVID-19 social restrictions [9,10,11]. Lockdown had an overall negative impact on movement, with reduced levels of physical activity seen across the world [9]. This is problematic because physical inactivity was already a global public health emergency, with an estimated 28% (1.4 billion people) of the global population being physically inactive [12]. Therefore, further reductions in physical activity is worrying. However, there were also differential effects, with some groups, such as those with long-term health conditions, e.g., hypertension, lung disease, and mental health problems being impacted more, leading to a reduction in physical activity [13] However, alongside this, there was some evidence that older adults were more likely to maintain their physical activity during lockdown [14,15]. 

Qualitative data are particularly useful in providing insights into peoples lived experiences during pandemics and other health emergencies, providing complimentary data to support epidemiological findings [16,17]. They can help untangle seemingly discrepant or surprising findings, as well as provide specific targets for intervention development. In order to inform intervention design, it is useful to map qualitative findings onto a theoretical intervention development framework [18]; one such framework is the Capability, Opportunity, Motivation, and Behaviour (COM-B) Model of Behaviour [19]. The COM-B posits that for a given behaviour such as physical activity to occur, there must be sufficient capability (physical or psychological), opportunity (physical or social environmental), and motivation (reflective or automatic) present. All three components are essential, if any of these components are weak or lacking, the behaviour has lower likelihood of occurrence [19]. The model also posits that both capability and opportunity influence motivation, making it the central mediator of the model [20]. 

During COVID-19 the COM-B model has been used to identify facilitators and barriers to behaviours such as adherence to COVID-19 social distancing guidelines in the UK [21,22]. Both studies identified a range of factors that contributed to compliance. Good social support and consistent, clear guidance were found to be drivers for compliance. From the COM-B model psychological capability, social opportunity and reflective motivation were important influences on compliance [23]. However, the COM-B model has been used less in relation to physical activity. In research conducted outside of the pandemic context, Howlett et al. (2019) found motivation to be strongly associated with physical activity in adults, with capability and opportunity partly mediating the association of physical activity behaviour [20]. Additionally, Willmott et al. (2021) identified, associations between behaviour, capability and opportunity through the mediating effect of motivation in young adults aged 18–35 years [23]. In research conducted into the COVID-19 pandemic, a review of 64 studies investigating correlates of physical activity behaviour identified that targeting capability on a psychological level and opportunity on a physical level may facilitate physical activity behaviour [24].

However, despite the large body of quantitative data on the impact of COVID-19 on physical activity, very few qualitative studies have been conducted. One American study aimed to understand 22 adults attempts to maintain an active lifestyle during the COVID-19 pandemic [25]. Participants described how during the initial lockdown they had a lack of motivation for exercise, however there was a gradual increase in motivation and participation in fitness exercises at home [25]. This study recruited only previous regular gym attenders, so findings may not be transferable to groups engaged in less intensive or less formalised physical activity. A qualitative study of 12 adults in Canada with a range of physical activity levels found that physical activity which could not be undertaken due to COVID-19 restrictions was often replaced by other types of activity such as online workouts, home exercise equipment, walking or biking on local paths. For those who required access to specific sports facilities some did not find an appropriate alternative activity [26]. However, due to the small sample size, this study did not allow for comparison across demographic groups. A UK qualitative study looked at barriers and facilitators to physical activity during COVID-19 and mapped to the COM-B model [27]. The study focused on those people that had been particularly affected by restrictions (younger adults aged 18–24, older adults aged 70+, parents with small children and those with physical or mental health conditions) and conducted in-depth interviews with 116 adults within these demographics. This study identified 4 themes (with seven subthemes): the importance of outdoor space, impact of COVID-19 restrictions, fear of contracting COVID-19 and level of engagement with home exercise. These were identified as either barriers or facilitators and mapped onto the COM-B model. Opportunity (physical) and motivation (reflective) were found to be factors influencing physical activity, no themes mapped to capability [27]. However, these studies remain relatively small in size [25], or focus on specific sub-populations [27]. We lack research providing broader insight into general public attitudes towards physical activity during COVID-19. Such research is important to capture how public opinion and behaviours are affected during a pandemic and to support planning to maintain physical activity, and thus physical health, for future health emergencies.

One method for capturing qualitative data from a large and diverse sample is via open-ended survey questions, free-text survey options give large numbers of participants the opportunity to explain their experiences in their own words, providing greater insight and detail into their lived experiences. This approach has been successfully used to further understand pandemic related barriers and concerns regarding catching COVID-19 of people with long-term respiratory conditions [28]. Therefore, the aim of the present study was to qualitatively explore the impact of the COVID-19 pandemic on physical activity in a large heterogeneous sample of people, and to map barriers and facilitators to the COM-B model to aid intervention development both during the COVID-19 pandemic and for future unforeseen circumstances in which individuals face challenging restrictions on their movement.

## 2. Method

### 2.1. Sample

Data were gathered from COVID-19 Social Study (CSS) participants. The CSS was a large scale, longitudinal, observational, panel study of adults living in the UK during the COVID-19 pandemic [29]. Survey inclusion criteria; aged ≥18 years, with a valid email address, and internet access. Participants of the study are not randomly selected and therefore not representative of the UK population, but the study employed a diverse recruitment strategy, including targeting marginalised and vulnerable groups, and contains a heterogeneous sample [29]. Data collection started in March 2020 via a weekly online questionnaire, which in September 2020 was reduced to monthly. The CSS predominantly involved quantitative survey questions designed to explore the impact of the pandemic on mental health and wellbeing, a one-off free-text module was introduced between 14 October and 26 November 2020 to gather more in-depth data on participants experiences during the pandemic. The COVID-19 restrictions in the UK during data collection are presented as a Appendix A. The module consisted of eight free-text questions about the impact of the pandemic on mental health, wellbeing, and coping methods; see Table 1 for the free text survey questions. The full study protocol is available at https://osf.io/jm8ra/, Accessed on 10 June 2022.

### 2.2. Data Extraction

A statistician working on the CSS generated a list of 29,303 individual words or numbers (of 3 or more characters) used in the free-text responses (including spelling mistakes). One researcher (VH), identified 304 of the 29,303 extracted words as relating to physical activity and generated a physical activity specific word list, see Appendix A. This list was used by the statistician to extract all free-text responses that contained one or more physical activity specific words for analysis. 21,260 quotes contained at least one physical activity specific word and were included in the first order coding. 7490 quotes from across all eight questions were coded as relating to physical activity. The 13,770 quotes excluded from the analysis used a key word, but not in relation to physical activity, e.g., “running the household and working from home has been challenging” or “juggling home schooling and working from home was exhausting”.

### 2.3. Data Analysis

Data were imported into an Excel spreadsheet. A thematic analysis approach was undertaken using an inductive approach following the six-step framework of Braun and Clark (2006) [30,31]. An inductive rather than deductive approach was taken as there were no preconceived themes expected to find in the data based on theory or pre-existing knowledge [32]. One researcher (VH) familiarised themselves with the data for each comment box and identified and developed potential codes that represented concepts discussed within each response. A second researcher (AF) familiarised themselves with a random selection of the dataset, 120 (0.5%) comments (15 quotes per question), and the two researchers discussed and agreed the coding system. Quotes were assigned first order codes, these were based on whether a participant spontaneously mentioned physical activity and whether the comment was positive/negative/mixed/neutral/no physical activity mentioned, see Table 2 for full definition. Comments which did not make any reference to physical activity were not included in further analysis.

Second order codes were then applied by VH to all remaining comments using the following structure:Type of physical activityChange in physical activity levels (increased/reduced/same/not clear)Barriers to physical activityFacilitators to physical activityImpact: mental health/wellbeing/physical health/none given

Appendix A provides examples of coding.

Once the coding was complete the responses were grouped according to first order coding (positive/negative/mixed/neutral) and the four groups reviewed individually. For the positive coding group, the facilitators (second order code) were reviewed, grouped and preliminary themes developed. For the negative coding group, the barriers (second order code) were reviewed, grouped and preliminary themes developed. For the mixed and neutral coding groups both facilitators and barriers were reviewed prior to developing preliminary themes. Preliminary themes from all four groups were reviewed together, common concepts and overlapping themes were identified, themes were then refined, reorganised and renamed where appropriated, e.g., the weather was identified as both a facilitator and barrier to physical activity, therefore it was reworded to ‘impacts’ physical activity. Themes were defined according to their overall impact on physical activity, discussed with co-authors and a final list agreed. Frequency of each theme was noted to give an indication of the prominence of different themes/subthemes.

Two researchers (VH, AF) independently mapped the themes onto the COM-B model [19]. Once complete the researchers compared their mapping. There was a high level of agreement between researchers.

## 3. Results

### 3.1. Demographics

25,051 individuals completed the survey during the study period containing the free text module. Response to the free-text questions was optional with 17,082 (68%) providing a response to at least one of the eight questions and 7490 quotes from 5396 participants (32%) mentioning physical activity.

Participant characteristics are presented in Table 3. The sample were predominantly female (84%), lived in England (79%), white (British/Irish/Other) (97%), degree educated (81%) and aged under 60 years old (57%). The subsample who mentioned physical activity differed from the overall sample: there were fewer people aged 46–59 years old, more women, more people with university education and more people with a physical health condition than the overall survey sample. Of the 7490 included quotes, 6636 (88.6%) were positive, 804 (10.7%) negative, 31 (0.4%) mixed and 19 (0.3%) neutral regarding physical activity. The most common forms of physical activity mentioned were ‘exercise’, followed closely by walking, then gardening, yoga and running.

### 3.2. Themes

Seven themes were identified from the data on the impact of the COVID-19 pandemic and associated restrictions on physical activity. A summary of themes and subthemes are presented in Table 4.

### 3.3. COM-B Model

The 7 themes were mapped onto the COM-B model, and categorised as either facilitators, barriers or both, see Table 4 for full details. Theme 4, perceived risks/threats to participation in physical activity, was the only theme identified as solely a barrier. Themes 5, 6 and 7 (importance of good health during COVID-19, use of technology to aid physical activity, importance of physical activity for mental health and wellbeing) were facilitators. Capability, opportunity, and motivation were all found to be drivers for behaviour change within these themes. The remaining themes (1, 2 & 3) were identified as both facilitators and barriers, with opportunity and motivation seen within the themes.

#### 3.3.1. Theme 1: The Importance of Outdoor Space for Physical Activity

The importance of outdoor space was the most frequently mentioned theme from 3167 participants. Access to a garden and greenspace was identified as having a positive impact on participation in physical activity. It allowed people to either, maintain regular activity, or engage in different activities during lockdown. Lack of access contributed to a reduction in physical activity. The desire to get outside to enjoy nature and outdoor space was identified as supporting many to be active, with walking outdoors a positive, enjoyable experience for many. Those who owned dogs highlighted the positive impact having to exercise their pet had on their own physical activity, with the additional benefit of social contact with other pet owners. Good weather and change in seasons affected outdoor physical activity, warm dry weather being associated with greater participation.

Access to garden or green space (1206 participants)

“Lots of gardening has kept me going” (Female, age 60+, England)

Desire for fresh air outdoors (780 participants)

“Exercise- getting out into nature has kept me sane” (Female, age 60+, Wales)

Dog ownership and social contact (498 participants)

“Having a dog is a fantastic outlet for me. From chatting to her and walking every day to meeting other dog walkers whilst out walking and I’ve been able to have socially distanced chat with them” (Female, age 60+, Wales)

Weather impacts the decision to participate in PA (111 participants)

“Trying to do more exercise but only achieve when weather was good” (Female, age 60+, England)

#### 3.3.2. Theme 2: Changes in Daily Routine Impacted Physical Activity

1164 participants had responses related to a change in their routine influencing physical activity. Some experienced a benefit, with increased time and flexibility to undertake physical activity, while others had increased pressures on routines, such as increased caring responsibility, which led to a reduction in activity. Changes in commuting habits had a large impact; positive quotes were identified from 899 participants related predominantly to the release of time normally spent commuting to focus on other activities. 77 participants comments were negative and linked to the loss of physical activity as part of their commute. A reduction in commuting habit was the most frequently mentioned facilitator for protecting mental health and wellbeing. Increases in caring responsibilities and home schooling had an overall negative impact on physical activity.

Changes in commuting habit impacted physical activity (1081 participants)

“Working from home means I don’t need to spend time travelling to work. I can use this extra time for other activities such as exercise” (Female, age 46–59, N Ireland)

“Not walking to work every day has had a big impact on my physical health–I have become a couch potato” (Female, age 60+, England)

Increase in caring responsibilities decreased physical activity (27 participants)

“My exercise is really low as I can’t get out to do things with children at home” (Female, age 46–59, England)

#### 3.3.3. Theme 3: COVID Restrictions Prevented Participation in Physical Activity

608 participants highlighted that the COVID-19 restrictions put in place by the Government prevented them from participation in their regular form of physical activity. This included activities such as going to the gym or travelling to participate in outdoor activities. All quotes were negative in relation to the impact of social distancing and travel restrictions with closure of facilities being the most commonly mentioned barrier having a negative effect on physical activity, mental health and wellbeing.

Closure of gyms and facilities (450 participants)

“Not being able to go to the gym or studio classes, which is what I would usually do to keep myself healthy and happy” (Female, age 18–29, England)

Impact of social distancing and travel restrictions (129 participants)

“I have had fewer opportunities to go walking with friends in the mountains which I normally do each week. This has led to reduced fitness and loneliness” (Female, age 60+, Wales)

#### 3.3.4. Theme 4: Perceived Risks/Threats to Participation in Physical Activity

A smaller but important theme was reported by 28 participants regarding perceived risk and threats to participation in physical activity. 23 respondents reported not feeling comfortable returning to previous indoor activities due to perceived risk of contracting the virus, even though some facilities had reopened. While being active outside was identified as an important factor for many, as the weather deteriorated with the approach of winter this raised a safety issue, 5 respondents reported a reduction in physical activity due to safety concerns around participating in the dark and a perceived reduction in safe spaces to exercise.

Concerns of catching COVID-19 while participating in physical activity (23 participants)

“I have stopped going to my exercise classes and I miss that, but when they did start up again, I didn’t want to go back due to the worry of catching the virus” (Female, age 60+, England)

Feelings of safety (5 participants)

“I don’t like going out on my own for walks and now its darker much earlier I’m not walking on an evening with my husband so rarely leave the house” (Female, age 46–59, England)

#### 3.3.5. Theme 5: The Importance of Protecting Physical Health during the Pandemic

339 participants mentioned the need to maintain or start physical activity to ensure they were in good health, with some participants specifically motivated to protect themselves against severe health consequences of contracting COVID-19.

“I have made my health a priority now so that if I did catch COVID I am in better shape to tackle it. I now exercise far more and am losing weight, this has had a positive impact on how I feel about myself” (Female, age 46–59, England)

#### 3.3.6. Theme 6: The Use of Technology to Aid Physical Activity

305 participants described using technology to support physical activity via apps and online platforms. 287 of the quotes were positive with regard to technology use for physical activity, including the ability to exercise when and where they wanted and the ease of use. 11 participants reported mixed experiences, predominantly because they missed the socialising and social contact during and after the activity.

“Positive impact has been an increase in exercise as zoom has enabled me to practice yoga and Pilates at times that suit me better from my spare room” (Female, age 46–59, England)

“I do yoga online but miss being able to do it in a studio with other people” (Female, age 60+, England)

#### 3.3.7. Theme 7: The Importance of Physical Activity for Mental Health and Wellbeing

Mental health and wellbeing were mentioned by 1022 (19%) participants in relation to physical activity. 585 quotes were related to the benefit the participants experienced on their mental health from physical activity. 16 participants specifically noted how important it was to maintain their physical activity throughout the pandemic to support their mental health and wellbeing. 421 quotes were negative, commenting that the loss or reduction in physical activity negatively impacted mental health and wellbeing. 88 participants specifically stated that the reduction in opportunities to do physical activity due to COVID-19 restrictions, social distancing restrictions and/or closures, had a negative impact on their mental health and wellbeing.

“I started exercising every day during the first lockdown and although I now only do 3–4 days a week, I think it helped my mental health” (Female, age 46–59, Wales)

“Not being able to go to the gym during lockdowns had an impact on my mental health” (Female, age 46–59, England)

## 4. Discussion

This study reports on findings from a large UK-wide survey of free-text data from 5396 adults and provides insight into the way the pandemic impacted daily lives, specifically the impact on physical activity. We identified a range of barriers and facilitators affecting physical activity, including the importance of outdoor space, changes in daily routine, COVID-19 restrictions, perceived risks or threats to participation, the importance of physical health, the importance of physical activity for mental health and the use of technology during the COVID-19 pandemic. Our findings show the majority had a positive experience of physical activity with 89% of respondents reporting either continuing, increasing activity, or specifying their enjoyment of being active. 18% of respondents mentioned the relationship between physical activity and mental health, with many specifically highlighting the benefits of physical activity to support their mental health. Others described instances where the restrictions meant they were unable to exercise as they would like and their concerns for the negative impact on their mental health.

The importance of greenspace, both access and the desire to be outdoors, was a main theme. This is not unexpected as exercise in a green space may help motivation to undertake physical activity by increasing enjoyment and escapism from everyday life [33]. Although the relationship between greenspace and physical activity is complex, pre-pandemic there was evidence that increased exposure and access may motivate people to be active and people enjoyed being active outdoors [34]. A study in Shanghai, China, of urban green space reported that green space is the preferred place for almost all types of outdoor physical activities [35]. Similarly, an observational study of 671 adults in Seattle, USA, demonstrated that participants who visited the park at least once over the week had increased minutes of daily physical activity than non-park users [36], suggesting that park visitation contributes to a more active lifestyle, although access alone is not responsible for the increase in physical activity [35]. Personal factors, such as having a companion, are significantly correlated with undertaking physical activity in green spaces, with availability of exercise equipment and picnic areas is also positively associated with frequency of physical activity [37]. Our study highlights the importance of being outdoors in a garden, greenspace, or countryside to get ‘fresh air’ with the majority reporting their enjoyment of exercising. Gardening and walking were the most frequently reported forms of physical activity in this theme. This highlights the importance of outdoor space to support physical activity and confirms that walking is an accessible and acceptable form of moderate physical activity for many. As a result, if we want to increase physical activity both generally and as a public health measure during future pandemics, current green spaces and parks need to be maintained as the quality, safety and accessibility affects their use for physical activity [37,38]. Developing green walkways in urban settings and maintaining footpaths in rural environments could provide a solution to an affordable and accessible form of physical activity across all ages and abilities.

Dog ownership was as a positive experience, with dog owners reporting they continued to exercise pets as before, thereby maintaining an active routine. Benefits were not just reported from the physical activity but also social interactions. These occurred spontaneously outdoors, often with other dog owners or people gardening. Social support has been associated with physical activity participation; being part of this social network potentially provided people with additional support to remain active prior to the pandemic [39]. Being outdoors meant they were able to follow COVID-19 guidelines, maintain social distancing and interact safely. Weather had both a positive and negative impact, with good weather encouraging people to go outdoors, while a change in season (going into winter) and related poorer weather were reasons for a reduction in activity. Seasonal changes in physical activity levels have been identified in previous research, with a drop seen during the colder, darker, winter months [40,41]. In order to increase physical activity during winter months and future pandemics, future interventions should include activities that can easily be undertaken both outdoors in greenspaces, and indoors, e.g., in the home environment to facilitate year-round participation. This corresponds with findings from a qualitative Canadian study [26], which found that people continued to be active if they were able to find alternative activities suitable for the lockdown environment such as online workouts or using home exercise equipment. Designing physical activity programs that can be used in both an indoor and outdoor setting could increase sustainability of use when outdoor access is restricted so that exercise routines are not disrupted by seasonality or during potential future social restrictions.

While changes in daily routine benefited some participants, particularly those who reduced their commute due to working from home, there were others who experienced a negative impact. Those who gained time from the loss of commute reported better mental health and using the time to participate in exercise and other healthy behaviours, e.g., cooking. However, while some participants expressed the intention to participate in more exercise, due to the this being a qualitative study we are unable to determine whether this translated to an increase in physical activity. The negative impact was described by those who lost an active commute and struggled to find a replacement, and those who experienced challenges of caring for young and older family members; with school and care facilities closed during lockdown this created additional time-related barriers to physical activity. Habit formation occurs when an action is consistently undertaken in the same context (time and place) [42] Changes in routine meant that the context which had facilitated the original action may no longer be in place, e.g., passing the gym on the way home from work. This may have made it harder to continue with previous habits, e.g., attending the gym, whilst also facilitating habits or patterns of inactivity being broken. Given, on average, it takes 66 days to form a new habit [42], the duration of each lockdown was sufficient to disrupt previous habits or to lead to the formation of new ones. Although many participated in the daily allowance of exercise during early lockdowns (i.e., going outdoors for exercise once a day), with the potential that this has formed a new ‘habit’ that may have persevered beyond lockdowns, the disruption of lockdowns does appear to have damaged previous routines for many individuals. As such, in future pandemic circumstances, targeted public health campaigns as lockdowns or social restrictions lift could help individuals to re-establish physical activity habits and routines or continue with newly created ones.

This study highlights a small but important theme of perceived risk to participation in physical activity, with participants noting a reduction in physical activity due to safety concerns, and a corresponding risk to mental health and wellbeing during a time of known stress. The potential of catching COVID-19 outweighed the known health benefits of returning to activities for some. The precautions put in place and the actions of other people did not adequately reduce the risk associated with interacting with others. The five participants who raised concerns about personal safety while participating in physical activity outdoors were female. With gyms and facilities closed and exercising outdoors encouraged, ongoing concerns about safety were highlighted. Staying safe during physical activity is just as important for health as the activity. Personal safety on the streets, public transport and around outdoor sports venues is essential to support ongoing participation. Ensuring high quality lighting and maintenance of streets, footpaths and greenspaces will encourage people’s feelings of safety, with long term benefits for future pandemic planning.

The importance of physical activity for mental health and wellbeing was identified across all themes. The positive effect on mental health was particularly evident in those who had access to outdoor space and the release of time spent commuting to participate in other activities. This echoes findings from quantitative studies during the pandemic [43]. Some participants took the opportunity to get more active during COVID-19, due to the beneficial effect of physical activity on mental health and wellbeing. Loss of access to gyms and social contact had an expected negative impact on physical activity which subsequently impacted mental health. Overall, this highlights that encouraging physical activity during COVID-19 was not just about maintaining physical health and public health, but also about supporting mental health. Indeed, there were multifaceted adverse effects on depression, anxiety, stress and wellbeing [44]. Given physical activity was evidently an important coping strategy for such psychological effects, this increases its importance as a public health measure for future pandemics.

The COM-B model of behaviour change illustrates potential mechanisms to determine which conditions need to be met to facilitate physical activity behaviour change at individual and population level [19]. Pre-pandemic, capability and opportunity were identified as influencing factors for physical activity, they also influence motivation, making it the central mediator of the model [20,23]. Motivation is a multidimensional construct and is a key factor that influences both initiation and maintenance of physical activity participation. It is thought that motivations for maintenance could be different from those that promoted individuals to make initial changes [45]. Motivation can also be different for different type of activity, age and gender in adults [46]. For example, there is evidence to support that in adults extrinsic motives (getting fitter, weight loss) dominate during the initiation of physical activity while intrinsic motives (competency and enjoyment) are important for maintenance [46,47]. A study looking at correlates of physical activity during COVID-19 identified *psychological* capability and *physical* opportunity as crucial for facilitating physical activity [24]. Theme 6, the use of technology to aid physical activity was the only theme that mapped to both capability (*physical* & *psychological)* and opportunity (*physical)* as a facilitator, suggesting that using technology as a means of engaging in physical activity was appreciated by participants. This could be of particular importance in future pandemics for those who have restricted access to in-person activities, e.g., those living in rural settings, those with mobility issues or those who would prefer to be active in the home setting. Some participants found the technology easy to navigate and enjoyed the flexibility, while others identified the loss of socialising and social contact as a negative; building in social time as part of the class could help resolve this. So it will be important to address these issues and concerns and build stronger, more accessible technological solutions to support physical activity ahead of future pandemics.

Themes found in this study are reflective of themes identified in an interview based qualitative study that looked specifically at those who were most likely to be impacted by the restrictions put in place during the pandemic [27]. Themes that overlapped with this study included, the importance of outdoor space, impact of COVID-19 restrictions, fear of contracting COVID-19, caring responsibilities, and using physical activity to protect mental health. Study authors mapped the themes predominately onto opportunity (physical) and motivation (reflective), while in this study themes were mapped across all three domains. While there are differences, the overlapping themes suggest that these were barriers for the broader population and not just those who might have been more affected by COVID-19 restrictions. The overlapping themes should be considered when developing physical activity interventions as we move out of this pandemic and into future pandemic planning.

A strength of this study is the large sample and the manual, structured approach to data analysis. Manual coding rather than using software allows for the researcher to gain a deeper understanding of the subject, to refine and interpret the data and gain a more nuanced picture. While this approach allowed the lead author to become familiar with the content, identify and apply the codes and themes, this may have led to researcher bias. Attempts to mitigate this included a second author (AF) checking a random selection of the coding. Limitations of the study are that although this was a large dataset it was not a randomly selected sample and therefore not representative of the UK population. The sample was predominantly female, white British and educated to degree level, which will limit the perspectives provided in the responses. Our study captures experiences mid pandemic and thus may miss the overall experience of physical activity during the pandemic. The survey questions did not specifically ask about physical activity, rather the survey collected data on the impact of the pandemic on mental health, wellbeing, and coping methods. Participants in this study were however only included if they spontaneously mentioned physical activity, 17,082 people responded to at least one of the free text questions of which 5396 mentioned physical activity. During the data collection period, Wales entered another lockdown, leading to different restrictions across the home nations. This had the potential to effect responses, during this time period the percentage of participants reporting a change in physical activity across the 4 home nations was very similar, therefore is unlikely to have affected the results. Access and availability of public infrastructure to support physical activity and socioeconomic position would limit the transferability of our results to other settings. Due to this being a qualitative study we are unable to identify associations between reported themes and impact on physical activity. Future work could investigate whether the positive and negative themes identified in this study were associated with quantitative measures of physical activity. Future research should consider seasonality, especially as the weather was identified as a barrier to outdoor activity. Moving forward, our findings supporting the benefits of physical activity for mental health and wellbeing and enjoyment of outdoor space can be used as evidence to support local and national public health initiatives that focus on the wider benefits of physical activity and the enjoyment of activity outdoors for mental health and wellbeing.

## 5. Conclusions

This study provides a large, novel, participant led, in-depth understanding of the experience of physical activity during a period of restrictions (lockdown and social distancing). It raises the importance of access to, and desire to be outdoors when active, highlighting the need to develop accessible, well maintained, greenspace in urban environments as a public health priority. Changes in daily routine had both positive and negative impacts on physical activity and moving forward, a flexible approach to working could support long term health benefits to both employers and employees. Concerns about catching COVID-19 and ongoing social distancing restrictions prevented some people from returning to activities. For some the concern of becoming seriously ill from COVID-19 encouraged them to become more aware of their health and become more active. Online resources were utilized to help support people to stay or become active. With the growing use of technology to aid participation in physical activity, this is an opportunity to develop a broad range of online/technology resources to promote and support physical activity as an alternative to conventional face to face delivery, providing a cost-effective and flexible way to deliver quality content to a large audience. Many people acknowledged and appreciated the positive impact of physical activity on mental health and wellbeing. Finally, while the closure of gyms had a negative impact on both physical activity and mental health, most participants reported enjoying physical activity, specifically walking outdoors. This study makes a number of recommendations for how to support physical activity during future pandemics. However, given the importance to global health of increasing physical activity levels, such measures also have a relevance to broader public health initiatives outside of health emergencies.

## Figures and Tables

**Table 1 ijerph-19-14784-t001:** COVID-19 Social Study Free text module questions.

Question Number	Question
1	Is there anything you would like to tell us about the changes that have been brought about by the COVID-19 pandemic and the impact these have had on your mental health or wellbeing?
2	What is bothering you the most about the pandemic? What aspects of it have you been finding most difficult?
3	Has the pandemic had any negative impacts on your mental health and wellbeing? If so could you tell us about these?
4	Has the pandemic had any positive impacts on your mental health and wellbeing? If so could you tell us about these?
5	How have your circumstances (e.g., work, housing, local area, finances, social networks, family life, responsibilities etc) contributed to your experiences (positive, negative or both) of the pandemic?
6	How have your personal attributes (e.g., age, gender, ethnicity, sexuality, health conditions etc) contributed to your experiences (positive, negative or both) of the pandemic?
7	What have been your methods for coping during the pandemic so far and which have been the most or least helpful?
8	Since the COVID-19 pandemic began, how have you been feeling about the future? What are you hopeful or concerned about?

**Table 2 ijerph-19-14784-t002:** Definition for First order coding.

Code Name	Numerical Coding	Definition
Positive	1	Statement indicated that they had enjoyed physical activity or had increased physical activity
Negative	2	Statement indicated that there had been a reduction in enjoyment in physical activity or had decreased physical activity
Mixed	3	Statement had both positive and negative elements
Neutral	4	Physical activity mentioned but no emotional response indicated or change in volume of physical activity.
Nil	5	Physical activity not mentioned

**Table 3 ijerph-19-14784-t003:** Demographic information of those who participated in data collection containing the free-text module (full sample) and those who mentioned physical activity within quotes (subsample).

Variable	Subsample * (Total = 5396)	%	Full Free Text Survey Sample (25,051)	%
**Gender**				
Female	4525	83.9	18,574	74.4
**Country**				
England	4257	78.9	20,056	80.1
Wales	764	14.2	3309	13.2
Scotland	334	6.2	1447	5.8
N Ireland	41	0.8	239	1
**Age group**				
18–29	235	4.4	1007	4
30–45	1160	21.5	4647	18.6
46–59	1698	21.5	8028	32.1
60+	2303	42.6	11,369	45.4
**Ethnicity**				
White (British/Irish/Other)	5214	96.6	24,110	96.6
**Education**				
Degree or above	4383	81.2	17,178	68.6
**Physical health condition**				
Yes	2068	38.3	7078	28.3
**Mental health condition**				
Yes	751	13.9	3749	15

* Subsample are those who mentioned physical activity within a quote.

**Table 4 ijerph-19-14784-t004:** Themes mapped onto COM-B model.

Theme	Facilitator	Barrier
**Theme 1: The importance of outdoor space for physical activity** Access to garden or green space Desire for fresh air outdoors Dog ownership and social contact Weather impacts the decision to participate in physical activity	Opportunity (*physical*) Motivation (*automatic*) & Opportunity (*physical*) Motivation (*physical*) & Opportunity (*social*) Motivation (*automatic*)	Opportunity (*physical*) Motivation (*automatic*) Motivation (*reflective*)
**Theme 2: Changes in daily routine impacted physical activity** Changes in commuting habit impacted physical activity Increase in caring responsibility decreased physical activity	Opportunity (*physical*)	Opportunity (*physical*) Opportunity (*physical*)
**Theme 3: COVID restriction prevented participation in physical activity** Impact of social distancing and travel restrictions Closure of gyms and facilities	Opportunity (*physical & social*)	Opportunity (*physical*) Opportunity (*physical*)
**Theme 4: Perceived risks/threats to participation in physical activity** Concerns of catching COVID-19 while participating in physical activity Feelings of safety		Motivation (*reflective*) Opportunity (*physical*) & Motivation (*reflective*)
**Theme 5: The importance of good physical health during COVID-19**	Motivation (*reflective*)	
**Theme 6: The use of technology to aid physical activity**	Capability (*physical & psychological*) Opportunity (*physical*)	
**Theme 7:** **The importance of physical activity for mental health and wellbeing**	Capability (*psychological*) Motivation (*automatic*)	

## Data Availability

The datasets generated and/or analysed during the current study are not publicly available due to stipulations set out by the ethics committee but are available from the corresponding author on reasonable request.

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
