# Peer review of "Physical Activity during the COVID-19 Pandemic in the UK: A Qualitative Analysis of Free-Text Survey Data"

_ijerph, 2022, doi:10.3390/ijerph192214784_

Round 1
Reviewer 1 Report
The introduction was presented systematically with support from a variety of current literature. The rationale of this study was well justified. The research design was appropriate with large data to support. The identified themes were discussed with relevant literature. However, the age group 18-29 obtained few subjects but it was fine with qualitative approach. Overall, the paper was well written and provided a good reference for the promotion of regular physical activity in order to tackle the pandemic.
Author Response
Thank you for the comments.
Reviewer 2 Report
Title
The title of the manuscript reads well and might spot interest in the reader.
Abstract
The abstract is well written and most of the important information is provided for the reader. My only concern is that the results could be more accurate.
Introduction
Authors provide a great overview of important concepts related to this study. Also, motivation is of important concept related to this study. Unfortunately, Authors do not introduce different forms of motivation. It is not meaningful to describe motivation as a general concept. Based on the self-determination theory, motivation ranges from extrinsic to intrinsic forms. It is important to distinguish between these forms of motivation because recent research has highlighted that it is specifically intrinsic motivation that is related to daily physical activity (Kalajas-Tilga et al., 2020), and not other forms of motivation.
Kalajas-Tilga, H., Koka, A., Hein, V., Tilga, H., & Raudsepp, L. (2020). Motivational processes in physical education and objectively measured physical activity among adolescents. Journal of Sport and Health Science, 9(5), 462–471. https://doi.org/10.1016/j.jshs.2019.06.001
Methods
Methods are well described.
Results
Results are described in a great detail.
Discussion
The discussion is well written, and Authors interpret the results from multiple angles. However, Authors miss several important nuances because they do not distinguish between different forms of motivation. Authors are recommended to study different forms of motivation based on the self-determination theory and then interpret the results how extrinsic and intrinsic forms of motivation might be related to physical activity.
Authors are recommended to provide practical implications of the current study.
Authors are recommended to provide more limitations and future directions of the current study.
Author Response
Thank you for your comments. We have amended as requested with the following added in response to each point.
Reviewer 2
- The abstract results could be more accurate.
Our response
and data from those who mentioned a physical activity related word in any context were included.
The sample were predominately female (84%), white (British/Irish/Other) (97%) and aged <60 years (57%).
Seven key themes were identified: the importance of outdoor space, changes in daily routine, COVID-19 restrictions prevented participation, perceived risks or threats to participation, the importance of physical health, the importance of physical activity for mental health and the use of technology.
- Authors are recommended to study different forms of motivation based on the self-determination theory and then interpret the results how extrinsic and intrinsic forms of motivation might be related to physical activity.
Our response
Motivation is a multidimensional construct and is a key factor that influences both initiation and maintenance of physical activity participation. It is thought that motivations for maintenance could be different from those that promoted individuals to make initial changes (Huffman 2020). Motivation can also be different for different type of activity, age and gender in adults (Molanorouzi 2015). For example, there is evidence to support that in adults extrinsic motives (getting fitter, weight loss) dominate during the initiation of physical activity while intrinsic motives (competency and enjoyment) are important for maintenance (Aaltonen 2012, Huffman 2020). – added to discussion
- Authors are recommended to provide practical implications of the current study.
Our response Moving forward, finding such as the benefits of physical activity for mental health and wellbeing and enjoyment of outdoor space can be used to support local and national public health initiatives that focus on the wider benefits of physical activity and the enjoyment of activity outdoors for mental health and wellbeing. – added to discussion
- Authors are recommended to provide more limitations and future directions of the current study.
Our response (limitations) The sample was predominantly female, white British and educated to degree level, which will limit the perspectives provided in the responses. Our study captures experiences mid pandemic and thus may miss the overall experience of physical activity during the pandemic.
Access and availability of public infrastructure to support physical activity and socioeconomic position would limit the transferability of our results to other settings.
Our response (future direction)
Future research should consider seasonality, especially as the weather was identified as a barrier to outdoor activity.
Reviewer 3 Report
Congratulations. Your study it's very interesting.
Which author did the authors follow for thematic analysis? Please explain your options.
Author Response
Thank you for highlighting the omission. The information has been added and sentence amended as requested.
- Which author did the authors follow for thematic analysis? Please explain your options.
Our response A thematic analysis approach was undertaken using an inductive approach following the six-step framework of Braun and Clark (2006). An inductive rather than deductive approach was taken as there were no preconceived themes expected to find in the data based on a theory of existing knowledge.
Round 2
Reviewer 2 Report
Authors have done extremely well job on revising the manuscript, I would like to congratulate Authors on that. My only concerns is that Authors could be more specific when describing the relations between different forms of motivation and objectively measured physical activity among adolescents. Specifically, previous research has clearly found that it is only intrinsic motivation that is related to daily physical activity among adolescents, and there was no association between extrinsic forms of motivation and daily physical activity (Kalajas-Tilga et al., 2020). I believe it is important a very information for practicing teachers.
Kalajas-Tilga, H., Koka, A., Hein, V., Tilga, H., & Raudsepp, L. (2020). Motivational processes in physical education and objectively measured physical activity among adolescents. Journal of Sport and Health Science, 9(5), 462–471. https://doi.org/10.1016/j.jshs.2019.06.001
Author Response
Thank you for your comments. Unfortunately this study only included adults participants (>18 years). It did not feel appropriate to include literature pertaining to adolescents as they are not part of the study population. A limitation of the study is a lack of objectively measured physical activity. The themes were generated from thematic analysis rather than actual physical activity data and mapped onto the COM-B model. Mapping to the COM-B model allowed for themes to be identified as potential areas for further research in the adult population.